# Concentration and Intakes of Macronutrients from Human Milk Do Not Differ by Infant Sex in Australian and Danish Cohorts

**DOI:** 10.3390/nu17233647

**Published:** 2025-11-21

**Authors:** Karina Dyrvig Honoré, Ching Tat Lai, Kim F. Michaelsen, Zoya Gridneva, Gitte Zachariassen, Donna T. Geddes

**Affiliations:** 1Department of Clinical Research, Faculty of Health Sciences, University of Southern Denmark, DK 5230 Odense, Denmark; gitte.zachariassen@rsyd.dk; 2Hans Christian Andersen Children’s Hospital, Odense University Hospital, DK 5000 Odense, Denmark; 3OPEN Patient Data Explorative Network, University of Southern Denmark, DK 5000 Odense, Denmark; 4School of Molecular Sciences, University of Western Australia, Crawley, WA 6009, Australia; ching-tat.lai@uwa.edu.au (C.T.L.); zoya.gridneva@uwa.edu.au (Z.G.); donna.geddes@uwa.edu.au (D.T.G.); 5ABREAST Network, Perth, WA 6000, Australia; 6UWA Centre for Human Lactation Research and Translation, Crawley, WA 6009, Australia; 7Department of Nutrition, Exercise and Sports, University of Copenhagen, DK 1958 Frederiksberg, Denmark; kfm@nexs.ku.dk

**Keywords:** sex-specific differences, human milk, macronutrients, 24 h milk intake, 24 h macronutrient intake, infants, lactation, breastfeeding

## Abstract

**Background/Objectives:** Human milk (HM) meets the nutritional needs of term-born infants. However, it remains unclear whether HM macronutrient concentrations and intakes differ by infant sex. We investigated sex- and country-specific differences in HM macronutrient and energy concentrations among exclusively breastfeeding Danish and Australian mothers, and sex-specific differences in 24 h HM intake and intakes of HM macronutrients, energy, and energy ratio in Australian infants. **Methods:** In this cross-sectional, multicenter study, 77 Danish and 84 Australian mothers donated HM samples between 2.5 and 5.5 months postpartum. Mid-infrared spectroscopy analyzed macronutrient concentration in the Danish samples. The creamatocrit method analyzed fat and biochemical assays analyzed lactose and protein in the Australian samples. Milk intake was measured using the test-weighing method. We used linear mixed-effect models to investigate sex- and country-specific differences. **Results:** There were no sex-specific differences in macronutrient and energy concentrations within either Danish or Australian cohorts. In all 161 HM samples, we found no sex-specific differences in lactose and protein concentrations. The 24 h median protein intake in Australian infants was 7.82 g in males and 7.26 g in females, (*p* = 0.274). The protein:energy ratio intake was 0.06 and 0.07 in male and females, respectively (*p* = 0.154). We also found no significant sex-specific differences in 24 h HM intake or intakes of fat, lactose, or energy. **Conclusions:** These findings suggest that males and females have similar macronutrient requirements during infancy. However, this needs to be confirmed in larger studies measuring 24 h milk intake.

## 1. Introduction

Human milk (HM) is uniquely composed to meet the term-born infant’s nutritional needs in the first 4–6 months of life, ensuring optimal growth and development [1]. HM macronutrient concentration changes over the course of lactation; however, after the dramatic shift during the establishment of lactation, which is completed by 6 weeks postpartum, lactose and fat concentrations remain relatively stable between 2 and 6 months of lactation, with protein concentration also remaining stable or gradually declining between 2 and 8 months postpartum [2,3,4,5,6,7]. Across 24 h, primarily fat concentration varies, while protein and lactose concentrations are relatively stable [6,8]. Additionally, fat concentration is lower pre-feed compared to post-feed [9]. It is known that factors such as gestational age, pre-pregnancy body mass index (BMI), current maternal body composition, and maternal age [2,4,10,11,12] can influence HM macronutrient concentration. However, it is still debated whether HM macronutrient concentration is influenced by infant sex [10,13,14], and if so, what the underlying mechanisms are. Sex-specific differences in milk macronutrient concentration were first observed in animal studies, where most studies found a higher milk yield and a higher macronutrient concentration in the milk for male offspring [15,16,17,18,19]. The animal studies raised the question of whether there could be a sex-specific difference in HM macronutrient concentrations.

Over the past 15 years, several human studies have investigated sex-specific differences in HM macronutrient concentration across populations worldwide [4,10,13,14,20,21,22,23,24,25,26]. However, the studies are conflicting and difficult to compare due to methodological differences that could impact the macronutrient concentration. Hence, there is no strong evidence for a sex-specific difference in HM macronutrients. It has been hypothesized that a sex-specific difference in HM macronutrient concentration could be biological compensation for the higher morbidity and mortality observed in male infants, referred to as the “male disadvantage” [27]. HM intake during established lactation has been reported to be 50–81 mL higher in male infants in several studies from different countries [28,29,30]; however, no studies have investigated whether macronutrient intake from HM during a 24 h period differs between male and female infants, which may be a more accurate approach than just measuring the concentration in a single milk sample when determining whether male and female infants have sex-specific macronutrient requirements.

Given these research gaps, this study aimed to investigate whether the fat, protein, lactose, and energy concentrations of HM from exclusively breastfeeding Danish and Australian mothers differed by infant sex and across countries, and to investigate whether there was a sex-specific difference in 24 h HM intake and intakes of macronutrients, energy, and the protein:energy ratio in exclusively breastfed Australian infants.

## 2. Materials and Methods

### 2.1. Participants and Human Milk Sampling

In this cross-sectional, multicenter study, we included data on healthy, non-smoking singleton mothers and their term-born infants from the Danish Odense Child Cohort (OCC) and two Western Australian cohorts, described elsewhere [31,32,33]. In Denmark, exclusively breastfeeding mothers were invited to donate a single HM sample at approximately three months postpartum when their infants were seen for their planned physical examination in the outpatient clinic. To ensure accurate recording of breastfeeding exclusivity, weekly SMS questionnaires were sent, starting at three days postpartum, with questions about the mothers exclusively breastfeeding and their use of infant formula and/or introduction to solids [34]. The Danish HM samples were collected between April 2012 and February 2014 between 8 am and 5 pm on the sampling day. Whether the samples were pre- or post-breastfeeding samples, from a full expression of the breast, collected manually or by a breast pump, was voluntary and not registered. The samples were stored at 5 °C and, within 3 days, frozen at −20 °C in 10 mL tubes (100 × 16 PP, Sarstedt, Nümbrecht, Germany) until analysis between 2 and 9 months after sample collection.

From the two Western Australian cohorts, exclusively breastfeeding mothers from the community were invited to donate HM samples between 1 and 6 months postpartum. Information on breastfeeding exclusivity was self-reported. The Australian HM samples were collected between January 2013 and December 2014 and from January 2018 to December 2020. The mothers were asked to select one breast from which they would donate the HM sample and were asked not to breastfeed or express milk from the selected breast for at least two hours prior to sample collection. Up to 20 mL or as much as possible was manually expressed directly into the sterile tubes (Greiner Bio-One, Kremsmünster, Austria). Details on the standardized sampling procedure are described elsewhere [35]. At 3 months postpartum, the Australian infants’ HM intake was calculated. The mothers weighed their infants before and after every breastfeeding session for 24 h using a digital scale at home (BabyWeighTM, Medela Inc., McHenry, IL, USA, resolution 2 g, accuracy ±0.034%) [28]. During the 24 h test-weighing session, a subgroup of mothers (*n* = 54) hand-expressed or pumped small (1–2 mL) pre- and post-breastfeeding HM samples at each breastfeeding session into 5 mL polypropylene plastic vials (Disposable Products, Adelaide, SA, Australia). The HM samples were labeled and stored at 4 °C in their home fridge freezer before being collected within 6–24 h and transported on ice to the laboratory at The University of Western Australia, where they were immediately aliquoted into sterile tubes (Sarstedt, Nümbrecht, Germany) and stored at −80 °C until analysis between 12 and 24 months after sample collection.

### 2.2. Macronutrient Analysis

The Danish HM samples were thawed at 5 °C overnight and in a 40 °C water bath afterward. They were homogenized using a sonicator (VCX 130, Sonics^®^, Sonics & Materials, Inc., Newtown, CT, USA). The macronutrient (fat, protein, and carbohydrate) concentrations were analyzed using mid-infrared transmission spectroscopy (Miris Human Milk Analyzer (serial no. 1303), Uppsala, Sweden), calibrated for this purpose. The carbohydrate concentration was determined based on the hydroxyl groups of carbohydrates, i.e., the total amount of carbohydrates measured. Approximately 70% of the carbohydrates measured by Miris HMA were lactose, and the remaining ones were HM oligosaccharides. The nutrient density was calculated based on the Atwater conversion: 4 kcal/g for protein and carbohydrates, and 9 kcal/g for fat. Kcal was converted to kJ by multiplying by 4.2. In a few of the Danish HM samples, milk volumes were insufficient, with only enough to analyze the protein concentration. The infants’ 24 h HM intake was not measured in the Danish cohort.

The Australian HM samples were thawed at 22 °C. The fat concentration was measured using the creamatocrit method [36]. Percentage fat was converted to fat concentration using the formulaFat (g/L) = 3.968 + (5.917 × creamatocrit %)

After measuring the fat concentration, the glass capillary was cut open to obtain the skim milk for protein and lactose analysis. Protein was measured in skim milk using the Bio-Rad DC Protein Assay Kit (Thane, India), adapted from the Lowry method [37], with HM protein standard prepared according to the Kjeldahl method [38]. Assay recovery was 98 ± 2% with a detection limit of 0.079 g/L. Lactose concentration was measured in skim milk using the Megazyme K-LOLAC Lactose Assay Kit. The HM intake in the Australian cohort was converted from grams to mL using the density of HM of 1.03 g/mL [39]. Since infants’ 24 h HM intake remains constant between 1 and 6 months postpartum [28] and protein and lactose concentrations have no significant circadian variation or changes within a breastfeed, the protein and lactose intakes were determined by multiplying the 24 h volume intake (L) by the measured protein (g/L) and lactose (g/L) concentrations in the single HM samples. The average fat concentrations (g/L) of each pre- and post-breastfeed sample were multiplied by the milk intake at each breastfeed during 24 h to account for the variability of fat concentration.

### 2.3. Covariates

In the Danish cohort, information on maternal age, gestational age, infant sex, and birth weight was obtained from medical records using the maternal individual’s identification number. Information on maternal pre-pregnancy BMI was collected from self-reported questionnaires during early pregnancy. Infant weight was measured using an electronic baby scale (Seca 717, Seca, Hamburg, Germany; accuracy ±2 g) on the day of the HM sample donation.

Among the Australian women, maternal age, gestational age, infant sex, and birth weight were self-reported. Maternal weight was measured using Seca electronic scales (±0.1 kg; Seca, Chino, CA, USA) at the time of milk sample collection. Height was self-reported by the mothers or measured against a marked wall. These values were used to calculate maternal BMI. Infant naked weight was measured on a digital scale (BabyWeighTM, Medela; accuracy ±2 g) at home when measuring 24 h HM intake.

### 2.4. Statistical Analysis

Descriptive statistics were used to describe the participating mother–infant pairs, the macronutrient concentration, the HM intake, and the intake of HM macronutrients, energy, and protein:energy ratio.

Data were presented as medians with interquartile range (IQR) if not normally distributed, visualized using histograms, and otherwise presented as mean ± standard deviation (SD).

Sex-specific differences in macronutrient and energy concentrations in the single samples were investigated using a linear mixed-effect model, including infant sex, country, maternal age, and maternal BMI as covariates and the interaction between infant sex and country. The sampling day was included as a random effect to account for the variance in HM sampling time postpartum. The same model was used to investigate sex-specific differences in 24 h HM intake and intakes of HM macronutrient, energy, and protein:energy ratio, including infant sex, maternal age, and maternal BMI as covariates for the potential effect on the dependent variable. The sampling day was included as a random effect. Gestational age was not included in the model because all the infants were born at term. The association between postpartum sampling day, macronutrients, and energy concentrations was investigated using a linear regression model. Data from the linear mixed-effect models and the linear regression model were presented as mean differences (β) with standard error (SE) and 95% confidence interval [CI]. The level of statistical significance was <0.05. The statistical analyses were performed in Stata/BE version 18.5 (StataCorp, College Station, TX, USA).

### 2.5. Ethics

The study complied with the Declaration of Helsinki II. The Danish part was approved by the Regional Committees on Health Research Ethics for Southern Denmark (ref. S-20090130, sub-protocols 12, 18, and 37) and the Danish Data Protection Agency (ref. 12/26892). The Australian part was approved by the Human Research Ethics Committee at The University of Western Australia (RA/4/1/2369, RA/1/4253, and RA/4/20/4023). Informed written content was obtained from all the participating mothers.

## 3. Results

### 3.1. Participant Characteristics

Across sites, a total of 161 mothers donated an HM sample. Twenty-four-hour HM intake and intakes of HM protein and lactose were calculated in 84 Australian infants, while fat, energy, and protein:energy ratio intakes were calculated in 54 Australian infants. The baseline characteristics of the mother–infant pairs are shown in Table 1. In total, 51 percent of the Danish and 54% of the Australian infants were males. The Danish HM samples were donated at 3.8 ± 0.7 months postpartum, while the Australian mothers donated HM samples at 3.3 ± 0.7 months postpartum. The maternal pre-pregnancy BMI was 22.7 (21.0–25.9) kg/m^2^ in the Danish mothers, and the maternal BMI at HM sampling was 24.0 (21.6–28.1) kg/m^2^ in the Australian mothers.

### 3.2. Macronutrient and Energy Concentrations in the Danish and Australian Human Milk Samples

There were no differences in HM macronutrient and energy concentrations by infant sex within the Danish or Australian cohorts (Table 2).

The protein and lactose concentrations did not differ between the countries (Table 3). However, the mean fat and energy concentrations in the Danish HM samples were lower than in the Australian HM samples (fat concentration: *β* = −13.88 (4.41), [−22.51; −5.24], *p* = 0.002; energy concentration: *β* = −472 (178), [−22.51; −5.24], *p* = 0.008). As this could be due to the different sampling techniques, no further sex-focused results for fat and energy concentrations are presented for the 161 samples (combined cohort). Sampling time postpartum was associated with significantly lower protein concentration (*β* = −0.91 (0.30), [−1.51; −0.32] *p* = 0.003), but not with lactose concentration (Table 3). Infant sex did not affect the protein and lactose concentration relationships between Denmark and Australia.

In the 161 Danish and Australian HM samples, protein concentration was 8.74 ± 3.14 g/L for male infants (*n* = 84) and 8.58 ± 2.36 g/L for female infants (*n* = 77). The lactose concentration when breastfeeding male (*n* = 81) and female (*n* = 72) infants was 75.10 ± 12.05 vs. 76.48 ± 13.04 g/L.

The box plots in Figure 1 show the median protein and lactose concentrations with IQR and outliers for Danish and Australian male and female infants. There was no significant difference in lactose concentration between HM for male and female infants (*β* = −4.03 (2.80), [−9.51; 1.45], *p* = 0.150). Additionally, we found no sex-specific differences in protein concentration (Table 3).

### 3.3. Human Milk, Macronutrient, Energy and Protein:energy Ratio Intakes in Australian Infants

The 24 h median (IQR) HM intake in Australian infants was 840 mL (726–970) in males (*n* = 45) and 756 mL (672–852) in females (*n* = 39) and was not significantly different (*p* = 0.052) (Table 4). The 24 h HM intake in mL/kg was also not significantly different (mean difference: 2.60 mL/kg; 95% CI: −13.25; 8.05; (*p* = 0.632)) (Table A1). The 24 h median (IQR) protein intake in Australian infants was 7.82 g (5.09–10.30) for males and 7.26 g (3.65–9.40) for females (*p* = 0.274), with a protein:energy ratio intake of 0.06 (0.05–0.7) in male and 0.07 (0.05–0.08) in female infants (*p* = 0.154). Furthermore, lactose, fat, and energy intakes did not differ by infant sex, with all *p*-values greater than 0.05 (Table 4). Greater maternal BMI at the time of HM sampling was associated with lower 24 h infant energy intake (β = −51.38 (−96.33; −6.42), *p* = 0.025) but not with 24 h HM or macronutrient intakes (Table A1).

## 4. Discussion

In this cross-sectional, multicenter study, we found no sex-specific differences in protein, lactose, fat, and energy concentrations in mature HM samples from exclusively breastfeeding Danish and Australian mothers. Additionally, we found no sex-specific differences in 24 h HM intake or intakes of macronutrients, energy, and protein:energy ratio in Australian infants. The fat and energy concentrations varied significantly across countries, likely explained by the use of different sampling techniques; however, we found no difference in lactose and protein concentrations between the Danish and Australian HM samples.

In line with our findings, no sex-specific differences in HM total protein concentration were previously reported [4,10,13,23]. Fischer Fumeaux et al. investigated this in milk samples collected from 27 mothers of preterm infants from birth to 4 months postpartum and from 34 mothers of term-born infants from birth to 2 months postpartum and found no sex-specific differences in protein concentration, either in the preterm or the term group [4]. A study from Poland that included milk samples collected from 77 mothers at four predefined 6 h periods during 24 h also reported no sex-specific differences in HM protein concentration. However, a sex-specific difference in fat concentration was reported by five studies [4,22,23,25,26]. In three of the studies, including 61, 50, and 25 HM samples, respectively, the fat concentration was significantly higher for male infants [4,22,26]. In contrast, one study including 71 HM samples found a higher fat concentration for female infants [23]. Finally, a Kenyan study, which included 81 samples, also found a higher fat concentration for female infants, but only if their mothers had a low socioeconomic status [25]. The conflicting results on sex-specific differences in HM fat concentration may be explained by the fact that existing studies do not account for the circadian variation and the increase in HM fat concentration with breast emptying during a breastfeed. Two of the studies did not report what time of the day the HM samples were collected [22,26], and one study collected HM samples three minutes after infant suckling, which may be at an intermediate stage of feeding, when fat concentration is known to increase progressively [23]. Therefore, previous findings of sex-specific differences in fat concentrations in single HM samples should be interpreted with caution.

The optimal method for collecting HM samples when analyzing macronutrient concentration is the collection of pre- and post-breastfeed HM samples for all feeds during a 24 h period to account for circadian and intra-feed variations, rather than relying on a single sample [40], which is a limitation in all the studies mentioned. To account for the variability in fat concentration in our study, the average fat concentration was calculated in 54 of the Australian samples by measuring fat concentration in all pre- and post-breastfeed samples expressed for 24 h. However, the 24 h collection method is demanding for the mother and could reduce the number of HM samples for study inclusion. To encourage more mothers to donate an HM sample, the HM sample collection in Denmark was not standardized. Differences in HM collection, storage, sample preparation, and analytical methods may, in part, explain the observed differences in fat and energy concentrations (which is largely driven by the fat concentration) between Denmark and Australia [41,42,43]. However, mid-infrared spectroscopy has been compared to laboratory methods and was found to be a reliable analytical method for measuring HM macronutrients [44,45]. Nevertheless, differences in how creamatocrit values (%) are converted to fat concentration may also lead to systematic variation. Earlier studies often used the Lucas method, which has been shown to overestimate fat concentration compared to mid-infrared spectroscopy [46]. However, in the Australian HM samples, the Meier method was used to interpret creamatocrit values, which tends to yield lower fat estimates than the Lucas method, but was found to be comparable to laboratory methods [36]. This suggests that the differences in fat and energy concentration between countries could be largely due to the sample collection techniques. Collecting a relatively large volume of milk (20 mL or more) from the breast or measuring and averaging fat concentrations in pre- and post-breastfeed samples (Australian cohort) may have resulted in higher values than in single samples collected at any time throughout breastfeeding (Danish cohort), as it is usually easier to collect samples before or early during the breastfeed. However, there were no sex-specific differences in fat or energy concentrations within both cohorts.

Regarding HM lactose concentration, and in contrast to our findings, a large study by Hahn et al., including 478 HM samples, found a lower odds ratio of 0.56 [0.36; 0.88] for a higher lactose concentration in HM for female infants [21]. However, samples were collected between 0 and 3 months postpartum (13.4 ± 8.13 days postpartum), which could represent all milk types—colostrum, transitional, or mature milk. Due to the increasing lactose concentration from colostrum to mature milk [3], comparing HM from different lactation stages may introduce bias if more HM samples from mothers of female infants were colostrum compared to samples from mothers of male infants. Yet, a study from Poland, including 77 mature HM samples pooled from pre- and post-breastfeed samples collected four times across 24 h at 4–8 weeks postpartum, also found higher odds of a higher carbohydrate concentration in HM for male infants [10]. Still, no strong evidence exists for a sex-specific difference in lactose and carbohydrate concentrations in HM. Furthermore, lactose concentration may increase with increasing milk production due to changes in the secretion of electrolytes contributing to the osmolarity [47,48], and the slightly higher carbohydrate concentration found in HM from mothers breastfeeding a male infants could be due to male infants’ higher HM intake, which was previously demonstrated in the Cambridge Baby Growth and Breastfeeding study measuring 24 h HM intakes in 94 infants [29]. In our study, the difference in mean 24 h HM intake in Australian male and female infants was 74 mL, which was not significant, supporting previous findings [20,49].

### Strengths and Limitations

The strengths of this larger-scale study included the standardized collection of HM samples in the Australian cohorts and the accurate estimate of fat intake by measuring fat concentration in pre- and post-breastfeed samples collected at every breastfeed during 24 h. The 24 h test weighing of the infants allowed us to estimate the 24 h HM intake, as well as the intakes of HM macronutrients and energy. Including HM samples from two countries allowed us to explore sex-specific differences in HM macronutrients across geography. To account for the changes in HM macronutrients over the course of lactation, the day of HM sampling postpartum was included in the comparison analyses as a random effect. However, only the protein concentration was significantly affected by postpartum sampling time. Our study setup would have been improved if the Danish HM samples had been collected using the same standardized protocol as the Australian mothers, with the 24 h test-weighing method performed at the Danish study site. Moreover, analyzing all the HM samples in the same laboratory using the same methods would have been preferred. Unfortunately, this was not possible due to ethical and legal conditions. The use of different sampling techniques, analytical methods, and the duration of freezer storage time could have contributed to differences in fat and energy concentrations between the Danish and Australian cohorts. Although the HM samples from Denmark and Australia were collected independently, both were intended for compositional analysis.

## 5. Conclusions

In this cross-sectional, multicenter study, we found no sex-specific differences in macronutrient and energy concentrations in mature human milk samples from exclusively breastfeeding Danish and Australian mothers. We observed a significant difference in fat and energy concentration across countries, which may be attributed to variations in HM sampling procedures. Furthermore, we observed no sex-specific differences in HM intake, as well as intake of HM macronutrients, energy, and protein:energy ratio, in Australian infants. These findings suggest that male and female infants have similar macronutrient requirements during infancy. However, our findings need to be confirmed in larger studies that measure 24 h human milk intake and use accurate estimates of infants’ 24 h macronutrient intakes.

## Figures and Tables

**Figure 1 nutrients-17-03647-f001:**
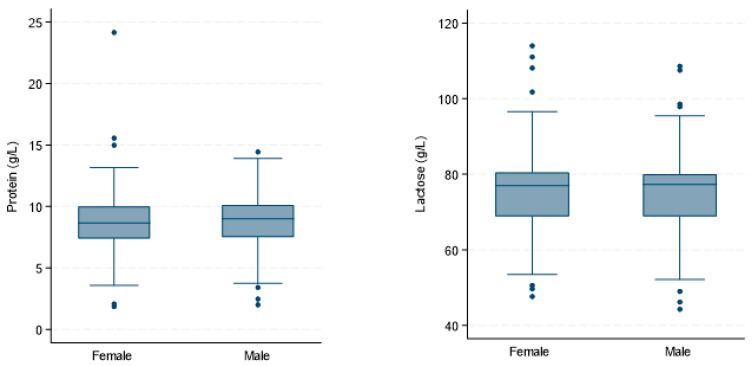
Absence of sex-specific differences in macronutrient concentrations in human milk (*n* = 161). The box plots show the protein and lactose concentrations (g/L) in human milk samples from exclusively breastfeeding Danish and Australian mothers.

**Table 1 nutrients-17-03647-t001:** Characteristics of the participating mother–infant pairs.

	Denmark	*n*	Australia	*n*
**Maternal age (years)**	30.3 ± 4.08 (21–39)	77	32.0 ± 4.9 (17–43)	80
**Maternal BMI (kg/m^2^) ^1^**	22.7 (21.0–25.9)	77	24.0 (21.6–28.1)	83
**Infant male sex % (male/female)**	50.7 (39/38)	77	53.6 (45/39)	84
**Gestational age (weeks)**	40.6 (39.6–41.1)	77	39.0 (38.0–40.0)	76
**Birth weight (kg)**	3.65 ± 0.50 (2.57–4.87)	77	3.48 ± 0.46 (2.44–4.71)	77
**Weight at sampling (kg)**	7.08 ± 0.99 (4.70−10.12)	77	6.45 ± 0.90 (4.93–9.51)	81
**Sampling time pp (months)**	3.8 ± 0.7 (2.6–5.4)	77	3.3 ± 0.7 (3.0–5.0)	84

Data are presented as mean ± standard deviation (SD) and range minimum–maximum (min-max) if they are normally distributed and as median with interquartile range (IQR) [median (Q1-Q3)] if they are not normally distributed. ^1^ The body mass index (BMI) for the Danish mothers was pre-pregnancy (BMI), while the BMI for the Australian mothers was measured at milk sampling.

**Table 2 nutrients-17-03647-t002:** Differences in HM macronutrient and energy concentration by infant sex within the Danish and the Australian cohorts.

		Denmark			Australia	
	*n*	*β* (SE), [95% CI]	*p*-Value	*n*	*β* (SE), [95% CI]	*p*-Value
**Protein g/L**	77	0.35 (0.28), [−0.20; 0.90]	0.212	79	0.23 (0.78), [−1.29; 1.76]	0.764
**Lactose g/L**	69	−1.79 (0.93), [−3.61; 0.03]	0.054	79	−4.13 (3.70), [−11.39; 3.12]	0.264
**Fat g/L**	69	−4.04 (3.22), [−10.35; 2.26]	0.209	79	4.70 (4.78), [−4.68; 14.07]	0.326
**Energy kJ/L**	69	−172 (128), [−424; 79.3]	0.180	79	237 (254), [−260; 735]	0.349

Data are presented with mean estimated difference (β) with standard error (SE), [95% confidence] interval, and corresponding *p*-values in macronutrient and energy concentrations in the Danish and Australian cohorts, separately. *p*-values are based on a linear mixed-effect-model, including sampling day postpartum as a random effect and infant sex, maternal age, and maternal BMI as fixed effects. Male sex is used as a reference in both the Danish and the Australian cohorts.

**Table 3 nutrients-17-03647-t003:** Differences in human milk macronutrient and energy concentrations by infant sex, country, and sampling time postpartum.

Component Concentration g/L	Sex (Male vs. Female)		Country (Denmark vs. Australia)		Sampling Time (Months)	
	*β* (SE), [95% CI]	*p * ^1^	*β* (SE), [95% CI]	*p * ^2^	*β* (SE), [95% CI]	*p * ^3^
**Protein (*n* = 161)**	0.22 (0.59), [−0.94; 1.38]	0.707	−0.68 (0.83), [−2.31; 0.95]	0.412	−0.91 (0.30), [−1.51; −0.32]	**0.003**
**Lactose (*n* = 153)**	−4.03 (2.80), [−9.51; 1.45]	0.150	−0.02 (3,38), [−6.64; 6.59]	0.994	−2.47 (1.44), [−5.32; 0.39]	0.090
**Fat (*n* = 153)**	NA	NA	−13.88 (4.41), [−22.51; −5.24]	**0.002**	−2.01 (2.12), [−6.20; 2.18]	0.344
**Energy (*n* = 153)**	NA	NA	−472 (178), [−820; −123]	**0.008**	−99 (85), [−268; 69]	0.250

Data are presented as mean difference (*β*), standard error (SE), and 95% confidence interval. ^1,2^ The *p*-values are based on a linear mixed-effect model, including sampling day postpartum as a random effect and infant sex, country, maternal age, and maternal body mass index as fixed effects, and the interaction between infant sex and country. ^3^ The *p*-values for the association between the dependent variables and the sampling day postpartum were based on a linear regression model. Bold font indicates a significant difference. NA—not available.

**Table 4 nutrients-17-03647-t004:** Sex-specific differences in 24 h human milk intake and intakes of macronutrient, energy, and protein:energy ratio in Australian infants.

24 h Intakes	*n*	Male	*n*	Female	*p*-Value ^1^
**Milk intake (mL)**	45	840 (726–970)	39	756 (672–852)	0.052
**Milk intake (mL/kg)**	43	129 (113–143)	38	126 (114–146)	0.632
**Protein intake (g)**	45	7.82 (5.09–10.30)	39	7.26 (3.65–9.40)	0.274
**Lactose intake (g)**	45	57.90 (50.39–72.72)	39	57.39 (45.83–68.68)	0.602
**Fat intake (g)**	29	42.18 (35.10–51.06)	25	39.84 (32.77–48.88)	0.265
**Energy intake (kJ)**	29	2476 (2123–3458)	25	2308 (1850–2788)	0.175
**Protein:energy ratio**	29	0.06 (0.05–0.07)	25	0.07 (0.05–0.08)	0.154

Since not all data were normally distributed, they are presented as medians and interquartile ranges. ^1^
*p*-values are based on a linear mixed-effect model, including sampling day postpartum as a random effect and infant sex, maternal age, and maternal body mass index as covariates.

## Data Availability

Restrictions apply to the availability of some or all data generated or analyzed during this study due to ethical reasons. The corresponding author will, on request, detail the restrictions and any conditions under which access to some data may be provided.

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
