# Peer review of "Concentration and Intakes of Macronutrients from Human Milk Do Not Differ by Infant Sex in Australian and Danish Cohorts"

_nutrients, 2025, doi:10.3390/nu17233647_

Round 1

Reviewer 1 Report

Comments and Suggestions for Authors

The topic is very relevant. We know that breast milk compounds adapt on how the infant is doing and growing. We know that fat content is higher for boys. We know that during a breastfeeding first low fat and then gradually higher fat content. I do see in table 2 that fat content is different between countries but not between male and female infants. Moreover in table 2 the sampling time (months) are different for protein content. That sampling moment is exactly that is not clear in whole publication. In abstract it is mentioned that HM samples were between 2.5 and 5.5 months postpartum. Why such a huge time difference? If this study was exactly designed for comparing countries and comparing male-female than this was more standardized. I now have the feeling that there were some samples and then they start analysing. It is also not clear how long the samples were in the freezer or for which purpose they were collected. I mostly strugle with the title as this was a not large sample size, so only in these samples they did not find difference between infant sex, which does not mean that there were not differences if the collection was more standardised. The title attracted my interest, but I know from other literature that these are differences between male-female within mothers, especially on fat content. I could also not find those references discussed in the discussion part. Further, line 28, should be added "beta" as not is looks like that the fat concentration was 13.88 g/L which is not true (see table 2). Moreover, I do not understand table 1 last row on sampling time pp (months). So for Denmark it was a mean of 3.8 months, but for Australia no mean and a percentage for 3 and 5 months. Therefor I really think that the collection of breast milk samples were completely different for different purpose. They should have looked for similar HM samples in Denmark at 3 months and only compare at 3 months with 3 months in Australia. If that was done, then such strong title could be used as statement that there were no differences in macronutrients between male-female. I would strongly recommended to give further clarification and adapt the title that only is these (strange) comparison of 2 countries sampling were performed.

Author Response

Comment 1: The topic is very relevant. We know that breast milk compounds adapt on how the infant is doing and growing. We know that fat content is higher for boys.

Response 1: Thank you for your comment. While several studies have suggested that breastmilk composition may adapt to the infant’s needs and that fat content could be higher for males, the evidence remains inconsistent. This inconsistency is largely due to heterogeneity in methodology and relatively small sample sizes, even in animal studies. That is why we also found it important to investigate whether the actual intake of macronutrients differed between infant sexes.

Comment 2: We know that during a breastfeeding first low fat and then gradually higher fat content. I do see in table 2 that fat content is different between countries but not between male and female infants.

Response 2: Thank you for your comment. You are right, fat concentration increases during breastfeeding. We found that fat concentration varied across countries, likely due to differences in sample collection, storage, and analytical methods. This underpins the importance of the analytical method used when assessing the concentration of components in human milk. However, we did not find a difference in the fat concentration according to infant sex. We additionally analyzed the macronutrient concentration separately in the Danish and Australian cohorts and found also no sex-specific differences. (Please see the table added)

Comment 3: Moreover in table 2 the sampling time (months) are different for protein content.

Response 3: Thank you for this observation. The analysis was based on time postpartum, not by grouping of earlier or later sampling times, and sampling day was included as a random effect in the mixed model. The average difference in sampling time between cohorts was approximately two weeks (3.3 vs 3.8 months), which falls within the 1-6 month period of established lactation, during which macronutrient concentration remain relatively stable. The column “Sampling time” shows the change in macronutrient concentrations in the milk samples over time. The protein concentration decreases significantly with increasing time postpartum, whereas changes in lactose, fat, and energy concentrations are not significant. This is consistent with previous literature showing that lactose and fat concentrations remain stable between 1-6 months of lactation, while protein concentration declines slightly.

  1. Khan, S.; Prime, D.K.; Hepworth, A.R.; Lai, C.T.; Trengove, N.J.; Hartmann, P.E. Investigation of short-term variations in term breast milk composition during repeated breast expression sessions. J. Hum. Lact. 2013, 29, 196–204.
  2. Mitoulas, L.R.; Kent, J.C.; Cox, D.B.; Owens, R.A.; Sherriff, J.L.; Hartmann, P.E. Variation in fat, lactose and protein in human milk over 24 h and throughout the first year of lactation. Br. J. Nutr. 2002, 88, 29–37.

Comment 4: That sampling moment is exactly that is not clear in whole publication. In abstract it is mentioned that HM samples were between 2.5 and 5.5 months postpartum. Why such a huge time difference? Response 4: Thank you for the question. In the Danish cohort, milk samples were collected at the scheduled 3-month follow-up visit. However, due to cancellations and rescheduling, some infants were slightly older and some were slightly younger at the time of the 3-month visit. In the Australian cohorts, milk samples were donated at 3 and 5 months of infant age. (See Table 1, last row; sampling time postpartum)

Comment 5: If this study was exactly designed for comparing countries and comparing male-female than this was more standardized. I now have the feeling that there were some samples and then they start analysing.

Response 5: Thank you for your comment. The manuscript was based on independent cohorts from two different countries, with HM samples collected for compositional analyses. The comparison between countries and between infant sex was conducted as a secondary analysis. Although the studies were not initially designed for direct cross-country comparison, both applied standardized analytical procedures and included detailed data, which strengthens the validity of the comparisons.

We believe it is a good ethical practice that, if you ask volunteer women to donate milk samples, researchers are obligated to make the most of them. Therefore, it is common practice to collect biological material, perhaps to investigate one or more specific questions, and later, as our knowledge advances, it is quite normal to want to study additional aspects using the material that has already been collected.

Comment 6: It is also not clear how long the samples were in the freezer or for which purpose they were collected.

Response 6: Thank you for your comment. The Danish milk samples were collected with the purpose of components analysis, together with, among others, blood-, hair-, urine-, and fecal samples, as well as anthropometric measures with the overall aim to explore nutrition, environmental factors, infections, and psychosocial influences on child health, growth, and development. The Australian milk samples were also collected with the purpose of component analysis. We have added the freezer storage time in the method section, which was between 2 – 9 months in the Danish cohort and 1-2 years in the Australian cohorts. The storage time has also been added as a limitation.

Comment 7: I mostly struggle with the title as this was a not large sample size, so only in these samples they did not find difference between infant sex, which does not mean that there were not differences if the collection was more standardised. The title attracted my interest, but I know from other literature that these are differences between male-female within mothers, especially on fat content.

Response 7: Thank you for this thoughtful comment. In the present study, we did not observe differences in either macronutrient concentrations or intakes by infant sex; however, we acknowledge that differences between infant sexes have been reported in other studies. We also acknowledge the limited sample size (although a total of 161 milk samples is not a small sample size either) and variability in collection methods. To better reflect this, we have changed the title to “Concentration and Intakes of Macronutrients from Human Milk do not differ by Infant Sex in Australian and Danish Cohorts.”

Comment 8: I could also not find those references discussed in the discussion part.

Response 8: Thank you for bringing this to our attention. We are sorry for that mistake. The empty reference 47 has been removed.

Comment 9: Further, line 28, should be added "beta" as not is looks like that the fat concentration was 13.88 g/L which is not true (see table 2).

Response 9: Thank you for your comment. We have added “beta” in the abstract.

Comment 10: Moreover, I do not understand table 1 last row on sampling time pp (months). So for Denmark it was a mean of 3.8 months, but for Australia no mean and a percentage for 3 and 5 months. –

Response 10: Thank you for your comment. We have added the mean for the Australian samples and included a percentage comment in the table footer.

Comment 11: Therefor I really think that the collection of breast milk samples were completely different for different purpose. They should have looked for similar HM samples in Denmark at 3 months and only compare at 3 months with 3 months in Australia. If that was done, then such strong title could be used as statement that there were no differences in macronutrients between male-female. I would strongly recommended to give further clarification and adapt the title that only is these (strange) comparison of 2 countries sampling were performed.

Response 11: We appreciate the reviewer's observation. Both the Danish and the Australian milk samples were collected for compositional analyses of human milk. However, we acknowledge that the sampling procedures were not identical between cohorts. We have added a statement to the limitation section to clarify this point, and we have also revised the Title.

Reviewer 2 Report

Comments and Suggestions for Authors

The authors compared the macronutrients of human milk by infant sex using two cohorts. Although the aim of the study is interesting some major concerns were raised about the methodology.

First of all, the two cohorts used different analytical methods for macronutrient determination. My first concern is that the authors didn’t compare the two methods using the same samples to find out, whether the results are comparable. Previous studies already proved, that these two different methods are not comparable, for example creamatocrit method prones to overestimate the lipid content compared to mid-infrared transmission spectroscopy (O'Neill EF, Radmacher PG, Sparks B, Adamkin DH. Creamatocrit analysis of human milk overestimates fat and energy content when compared to a human milk analyzer using mid-infrared spectroscopy. J Pediatr Gastroenterol Nutr. 2013 May;56(5):569-72. doi: 10.1097/MPG.0b013e31828390e4), but the other macronutrients can also be significantly different (Silvestre D, Fraga M, Gormaz M, Torres E, Vento M. Comparison of mid-infrared transmission spectroscopy with biochemical methods for the determination of macronutrients in human milk. Matern Child Nutr. 2014 Jul;10(3):373-82. doi: 10.1111/j.1740-8709.2012.00431.x).

My second major concern is that although you didn’t compare the results measured with the two different methods, you pooled the results of the two cohorts (although thy are not comparable) and divided into boys and girls. The results of this comparison is not reliable, and statistical comparison of these two groups has no sense.

The samples of the two cohorts were also stored differently, in Denmark at -20°C, while in Australia at -80°C and there is no information, after how much months/years were samples analysed. Storage time might also affect the macronutrient composition and different temperatures might affect differently the degradation.

Author Response

The authors compared the macronutrients of human milk by infant sex using two cohorts. Although the aim of the study is interesting some major concerns were raised about the methodology.

Comment 1: First of all, the two cohorts used different analytical methods for macronutrient determination. My first concern is that the authors didn’t compare the two methods using the same samples to find out, whether the results are comparable.

Response 1: We thank the reviewer for this important comment. We agree that the two cohorts were analyzed using different analytical methods for macronutrient determination. Unfortunately, we did not have the opportunity to ship the Danish samples to Australia or the Australian samples to Denmark for analysis. This has been acknowledged as a limitation in the manuscript.

Comment 2: Previous studies already proved, that these two different methods are not comparable, for example creamatocrit method prone to overestimate the lipid content compared to mid-infrared transmission spectroscopy (O'Neill EF, Radmacher PG, Sparks B, Adamkin DH. Creamatocrit analysis of human milk overestimates fat and energy content when compared to a human milk analyzer using mid-infrared spectroscopy. J Pediatr Gastroenterol Nutr. 2013 May;56(5):569-72. doi: 10.1097/MPG.0b013e31828390e4), but the other macronutrients can also be significantly different (Silvestre D, Fraga M, Gormaz M, Torres E, Vento M. Comparison of mid-infrared transmission spectroscopy with biochemical methods for the determination of macronutrients in human milk. Matern Child Nutr. 2014 Jul;10(3):373-82. doi: 10.1111/j.1740-8709.2012.00431.x).

Response 2: Thank you for your comment, which we agree on. We have already mentioned this in the Discussion, including the citation. The paper cited refers to the Lucas method, which calculates higher values than the Meier method. We used the Meier method. However, we have added as a limitation that the use of different analytical methods could have systematically affected both the fat, protein, and lactose content.

Comment 3: My second major concern is that although you didn’t compare the results measured with the two different methods, you pooled the results of the two cohorts (although thy are not comparable) and divided into boys and girls. The results of this comparison is not reliable, and statistical comparison of these two groups has no sense.

Response 3: Thank you for bringing this concern to our attention. Indeed, direct pooling of already collected data in a post hoc manner across sites can be problematic due to differences in the distributions of one or more measurements. However, pooling cohorts to increase sample size is a standard procedure if methodology is considered comparable and often used in meta-analysis. The Country was included as a covariate. Only concentration results were pooled, with more biologically meaningful results (intakes) tested within the Australian sample only.

We have additionally analyzed the sex-specific differences in the macronutrient concentration separately in the Danish and Australian cohorts, and found no sex-specific differences. Thus, there is no opposite “lumping” effect of pooling. (See the table below with results). We have added the Table in Appendix.

                Denmark

                 (male used as reference)

Australia

(male used as reference)

n

Β (SE), [95% CI]

P

n

Β (SE), [95% CI]

P

Protein g/L

77

0.35 (0.28), [-0.20; 0.90]

0.212

79

0.23 (0.78), [-1.29; 1.76]

0.764

Lactose g/L

69

-1.79 (0.93), [-3.61; 0.03]

0.054

79

-4.13 (3.70), [-11.39; 3.12]

0.264

Fat g/L

69

-4.04 (3.22), [-10.35; 2.26]

0.209

79

4.70 (4.78), [-4.68; 14.07]

0.326

Energy kJ/L

69

-172 (128), [-424; 79.3]

0.180

79

237 (254), [-260; 735]

0.349

P-values are based on a linear mixed effect-model, including sampling day postpartum as a random effect, infant sex, maternal age, and maternal BMI as fixed effects.

Comment 4: The samples of the two cohorts were also stored differently, in Denmark at -20°C, while in Australia at -80°C and there is no information, after how much months/years were samples analysed. Storage time might also affect the macronutrient composition and different temperatures might affect differently the degradation.

Response 4: Thank you for the comment. We agree that different storage times and temperatures could affect macronutrient concentrations differently across countries. We have elaborated on this in the Discussion and as a limitation, and we have added the freezer storage time for the Danish and the Australian samples, respectively, in the Method section

Reviewer 3 Report

Comments and Suggestions for Authors

In the introduction the authors state that "also infant sex could influence HM macronutrient concentration. 49" please provide an explanation on how the infants sex influences human milk or remove the line. 

The 47th reference is missing.  

In the discussion the authors state "In line with our findings, no sex-specific differences in HM total protein concentra- 251
tion were previously reported [4, 8, 19, 20]. " this should be expanded a bit more.

Also, the authors did not comment on their findings on the results of the difference in energy (kJ/L) concentrations, therefore this should be added in the discussion.

In the results I suggest removing the "may" from the line "These findings suggest that male and female infants may have similar mac- 328
ronutrient requirements during infancy." since the authors did not find any difference should be confident in their findings. 

Author Response

Comment 1: In the introduction the authors state that "also infant sex could influence HM macronutrient concentration. 49" please provide an explanation on how the infants sex influences human milk or remove the line.

Response 1: Thank you for your suggestion. We have added that it is debatable whether infant sex influences human milk macronutrient concentration, and added two references. Two studies showing that infant sex influences HM macronutrient content, and one study showing no effect of infant sex on HM macronutrient content. Further, we have added that the mechanism is unclear if infant sex influences HM macronutrient concentration.

Comment 2: The 47th reference is missing.  

Response 2: Thank you for pointing this out. This was an error, and the reference 47 has been removed.

Comment 3: In the discussion the authors state "In line with our findings, no sex-specific differences in HM total protein concentration were previously reported [4, 8, 19, 20]. " this should be expanded a bit more, -

Response 3: Thank you for the advice. We have added more details of the findings in each of the 4 studies.

Comment 4: Also, the authors did not comment on their findings on the results of the difference in energy (kJ/L) concentrations, therefore this should be added in the discussion.

Response 4: Thank you for your suggestion. We have already mentioned the findings of a difference in energy concentration across countries, but we have also noted that the fat concentration largely drives the energy concentration.

Comment 5: In the results I suggest removing the "may" from the line "These findings suggest that male and female infants may have similar macronutrient requirements during infancy." since the authors did not find any difference should be confident in their findings.

Response 5: Thank you for the advice. We have removed “may” in that line in the conclusion.

Round 2

Reviewer 1 Report

Comments and Suggestions for Authors

It is an assumption that during the 1-6 month period of established lactation, the macronutrient concentration remain relatively stable. the current paper will not enlight on this topic unfortunately. Therefor I still think that the title is not correct. In the response letter they explained that the Denmark cohort should have given HM sample at 3 months. Why included samples that were done later? Same for Australian cohort, only include the samples at 3 months. Then do the analysis again. The title should be at 3 months postpartum. Then you can compare fat content between boys and girls, not with the current collection of samples.  

Author Response

Dear Reviewer,

Thank you for reading our manuscript once again and for your comments.

Changes in the manuscript in the first round are highlighted in yellow, and changes after the second round are shown using track changes.

Comment 1: It is an assumption that during the 1-6 month period of established lactation, the macronutrient concentration remain relatively stable. the current paper will not enlight on this topic unfortunately.

Response 1: We thank the reviewer for this comment; however, we disagree that it is an assumption. It is been demonstrated in many studies and is well accepted, but we do acknowledge that it is stated unclear in the introduction. We have elaborated on that and added 3 references.

Comment 2: Therefor I still think that the title is not correct.

Response 2: Thank you for your comment. We have acknowledged that and the title has been changed with cohorts specified: “Concentration and Intakes of Macronutrients from Human Milk do not differ by Infant Sex in Australian and Danish Cohorts

Comment 3: In the response letter they explained that the Denmark cohort should have given HM sample at 3 months. Why included samples that were done later? Same for Australian cohort, only include the samples at 3 months. Then do the analysis again. The title should be at 3 months postpartum. Then you can compare fat content between boys and girls, not with the current collection of samples. 

Response 3: Thank you for your comment. In the Danish cohort, the human milk sampling was scheduled for the day the infant attended the outpatient clinic for the routine physical examination. This first physical examination was predetermined to take place at 3 months of age. However, because approximately 2,300 infants were to be scheduled for the 3-month visit (and only two staff members carried out all the examinations), and because some families were unable to attend on their originally assigned date, it was not feasible to see all infants exactly on the day they turned 3 months old. There was only a small subgroup of mothers who donated milk samples, and if we were to only include samples donated at exactly 3 months, we would end up with very few samples.

However, we disagree with you. As per the above answer, human milk composition does not change significantly within the sampling period between 2.5 and 5.5 months, with a mean sampling time of 3.8 (Denmark) and 3.3 months (Australia); thus, the sampling time is acceptable. Additionally, we have accounted for sampling time in the models. Table 1 has been updated, and the means and ranges of sampling time are similar between the cohorts. No further changes were made in the manuscript.

Reviewer 2 Report

Comments and Suggestions for Authors

Thank you for answering my questions, but I still believe that due to the very different methodologies used, the data from the two cohorts cannot really be combined, as there are too many unknown variables.

For example, the significant difference in fat and energy content between the two cohorts could be due to several reasons:

  • analytical reasons (different methods used for the determination)
  • a natural difference between the two nations (e.g.: effect of different maternal diet, genetics or other maternal factors)
  • different sampling times (before or after breastfeeding, during the day when: circadian variation within a day).

Since none of these reasons can be ruled out, there are too many unknown factors for the data to be reliably combined and it is not enough to mention this fact in the Limitations section.

You mentioned that data are often combined in meta-analyses to increase the number of subjects, but in such cases it is important that the two different methodologies are comparable. In meta-analyses, in order to compare different methodologies, we perform a subgroup analysis to determine how much the values determined by the two methods differ, i.e., whether the data can be combined.

In the present case, due to the significant difference, we may obtain less biased results if we examine the data of the two cohorts separately according to the infants’ sex than if we examine the two cohorts together.

Author Response

Dear Reviewer 2,

Thank you for reading our manuscript once again and for the concerns you have raised.

Changes in the manuscript in the first round are highlighted in yellow, and changes after the second round are shown using track changes.

Comment 1: Thank you for answering my questions, but I still believe that due to the very different methodologies used, the data from the two cohorts cannot really be combined, as there are too many unknown variables.

For example, the significant difference in fat and energy content between the two cohorts could be due to several reasons:

  • analytical reasons (different methods used for the determination)
  • a natural difference between the two nations (e.g.: effect of different maternal diet, genetics or other maternal factors)
  • different sampling times (before or after breastfeeding, during the day when: circadian variation within a day

Since none of these reasons can be ruled out, there are too many unknown factors for the data to be reliably combined and it is not enough to mention this fact in the Limitations section.

You mentioned that data are often combined in meta-analyses to increase the number of subjects, but in such cases it is important that the two different methodologies are comparable. In meta-analyses, in order to compare different methodologies, we perform a subgroup analysis to determine how much the values determined by the two methods differ, i.e., whether the data can be combined.

In the present case, due to the significant difference, we may obtain less biased results if we examine the data of the two cohorts separately according to the infants’ sex than if we examine the two cohorts together.

Response 1: We agree with the Reviewer’s concerns. Whilst we believe that the used measurement methods for macronutrients could be comparable, the differences in sampling technique could be the reason for differences in fat concentration between the cohorts, as collecting a relatively large volume of milk (20 mL or more) from the breast, or measuring and averaging fat concentrations in pre- and post-breastfeed samples in Australian cohort may result in higher values than in single samples collected at any time throughout breastfeeding (Danish cohort), as it is usually easier to collect samples before or just after starting the breastfeed than after it.

Considering this, we have removed the results for fat and energy for the combined cohort and are only discussing fat results within the cohorts. This speculation has been added to the Discussion.